# LAP: Liability Antibody Profiler by sequence & structural mapping of natural and therapeutic antibodies

**Tadeusz Satława**[1], **Mateusz Tarkowski**[1], **Sonia Wróbel**[1], **Paweł Dudzic**[1], **Tomasz Gawłowski**[1], **Tomasz Klaus**[2], **Marek Orłowski**[2,3], **Anna Kostyn**[2], **Sandeep Kumar**[4], **Andrew Buchanan**[5], **Konrad Krawczyk**[1]*

1 Natural Antibody, Kraków, Poland, 2 Pure Biologics, Wrocław, Poland, 3 Department of Biochemistry, Molecular Biology and Biotechnology, Faculty of Chemistry, Wrocław University of Science and Technology, Wrocław, Poland, 4 Moderna Inc, Cambridge, Massachusetts, United States of America, 5 Biologics Engineering, AstraZeneca, Cambridge, United Kingdom

* konrad@naturalantibody.com

**Data Availability Statement:** All relevant data are within the manuscript and its Supplementary Information files. The LAP application is available at lap.naturalantibody.com.

## Abstract

Antibody-based therapeutics must not undergo chemical modifications that would impair their efficacy or hinder their developability. A commonly used technique to de-risk lead biotherapeutic candidates annotates chemical liability motifs on their sequence. By analyzing sequences from all major sources of data (therapeutics, patents, GenBank, literature, and next-generation sequencing outputs), we find that almost all antibodies contain an average of 3–4 such liability motifs in their paratopes, irrespective of the source dataset. This is in line with the common wisdom that liability motif annotation is over-predictive. Therefore, we have compiled three computational flags to prioritize liability motifs for removal from lead drug candidates: 1. germline, to reflect naturally occurring motifs, 2. therapeutic, reflecting chemical liability motifs found in therapeutic antibodies, and 3. surface, indicative of structural accessibility for chemical modification. We show that these flags annotate approximately 60% of liability motifs as benign, that is, the flagged liabilities have a smaller probability of undergoing degradation as benchmarked on two experimental datasets covering deamidation, isomerization, and oxidation. We combined the liability detection and flags into a tool called Liability Antibody Profiler (LAP), publicly available at lap.naturalantibody.com. We anticipate that LAP will save time and effort in de-risking therapeutic molecules.

## Introduction

Antibodies are glycoproteins produced by the immune system to identify and neutralize foreign pathogens such as viruses, bacteria, and other infectious agents. The protective function of antibodies has been harnessed for therapeutic purposes, particularly to selectively target cancer cells. There are currently more than 160 monoclonal antibody drugs approved for use in the world, with many more in development in a market estimated to be worth close to $200 billion [1].

**Funding:** The author(s) received no specific funding for this work.

**Competing interests:** I have read the journal's policy and the authors of this manuscript have the following competing interests: TS, SW, PD, TG are employees of NaturalAntibody. TK, MO, AK are employees of Pure Biologics. AB is an employee of AstraZeneca. SK is an employee of Moderna, Inc. and may hold stocks/stock options in this and other biopharma companies.

In addition to their therapeutic efficacy, the developability of antibodies is an essential consideration in their production and clinical use [2,3]. Developability refers to a set of characteristics that determine whether an antibody is suitable for large-scale production, formulation, long-term storage, and administration. These characteristics include factors such as stability, solubility, polyreactivity, ease of manufacturing, and physicochemical integrity of the drug product during manufacturing, shipping, etc. Antibodies that possess favorable developability properties are expected to be successfully developed and brought to market [4]. Moreover, they are more likely to have a long shelf-life, reduced manufacturing costs, and improved patient outcomes [4].

A critical step in the development of therapeutic antibody products is the precise definition and predictive characterization of their amino acid sequences and structures. The slightest oversight in the annotation of sequences, or identification of mutations or residues that may be important in the stability and efficiency of antibodies could lead to a reduction in the effectiveness of any promising lead candidates [5]. Thus, it is necessary to create highly effective and precise bioinformatics tools to ensure the development of antibody products as safe and potent medicines with minimal drug product heterogeneity [6].

Though there is a range of computational methods for de-risking the developability of antibodies [2,6–8], one of the staple early methods remains the identification of 'sequence liabilities' [9]. Undesirable chemical modifications can be associated with a range of sequence motifs that can be two or more amino acid residues long, or be associated with specific amino acids (e.g. Met, Trp oxidation, Asn, deamidation, Asp, Isomerization, and so on). Such liabilities include, among others, factors such as glycosylation [10], deamidation [11] isomerization [12], and more. By identifying such sequence liabilities early in the drug development process, scientists can design and engineer antibodies with improved developability and efficacy, increasing chances for desirable drug function (antigen recognition or effector function) or good stability of the product during manufacturing, shipping, and storage (shelf-life).

It is well known that the identification of sequence liabilities purely based on the sequence motifs is over-predictive [9]. Physical or chemical reaction mechanisms, rate, and structural information such as solvent exposure provide us with the key information on the severity of these liabilities. Not all susceptible motifs will undergo a chemical reaction [7]. Structural conformation [13] or chemical conditions such as pH stress [7] all impact the rate at which the sequence motifs will undergo modifications. There were multiple efforts to predict the rate of degradation of certain motifs via computation [13–15]. Such efforts produce good results, however, they are still hampered by small datasets and applicable only to a select set of modifications (e.g. in vitro liabilities only).

Here, we have undertaken a survey of sequence liabilities from major sources of information from the public domain, namely therapeutics, patents, natural antibodies, GenBank, and literature. We provide occurrence statistics for sequence liabilities in antibodies sourced naturally as well as those destined for therapeutic applications. Our results show that sequence liabilities have very high occurrences within all datasets, which is in line with previous information that most such liabilities are highly over-predictive. Therefore, methods to reduce the number of false positives are desirable to focus engineering efforts on the liabilities that can carry higher risk.

To address the over-predictability issue, we have benchmarked computational methods to ignore certain sequence liabilities, based on germline signature, success in clinical trials, and structural information. We demonstrate that such flags reduce the significant amount of all liabilities detected by simple sequence analysis and they find correlation with experimental datasets. For convenience, we combine the methods into a single online tool, LAP (https://lap.naturalantibody.com), which we hope will facilitate the identification of liabilities that pose actual risk of chemical degradation.

## Results

### Mapping sequence liability prevalence in major sources of antibody information

We defined an *Antibody Liability Reference* (Table 1) based on a compilation of the motifs described in the literature and definitions used in pharmaceutical and biotechnology companies (see Methods). Such liabilities are detected employing sequence information alone, so it is fundamental to investigate their prevalence across globally available antibody datasets. For this purpose, we studied the frequency of liabilities across a large dataset of antibody sequences, sourced across the natural and therapeutic spectrum. We have defined deamidation with 3 levels of severity (high, medium, and low) and fragmentation with 2 levels (high and medium); others have only 1 severity level.

For each heavy and light sequence in our datasets, we applied our Antibody Liability Reference. Each sequence where any liability was detected (e.g. isomerization), was marked as having a sequence liability and we noted the total number of such motifs, their type, and severity in each sequence. The number of all sequences we detected across our datasets, as having any sequence liabilities is given in Fig 1. Hardly any sequences are free of sequence liabilities, with the lowest percentage marked at 89.8% of light chain from the literature. There does not appear to be a marked difference between liability frequency between human and non-human sequences. A number of sequence liabilities associated with a therapeutic do not appear to be strongly distinct between molecules that are already marketed and those still in clinical trials. The marketed antibodies have a slightly higher percentage of sequences with liabilities (96.8%) rather than all therapeutics (96.1%), which could be indicative of the progress in the field. Testing the statistical difference in the number of liabilities between the different datasets is sometimes insignificant (please see S1 Fig).

We calculated how many liabilities one could expect per sequence in any of our datasets (Fig 2). The mean number of liabilities per sequence does not appear to be strongly associated with any dataset, with single sequence datasets having means in the range of 3–3.4 liabilities per sequence and 5.8–6.5 liabilities per molecule in the case of paired datasets (Table 2). There does not appear to be a staggering difference between organism sources of a sequence considering the mean number of liabilities (Table 2). As before, marketed therapeutics have a slightly higher number of liabilities (6.13) than all therapeutics (5.85), which could be the effect of newer therapeutics accumulating more developability product knowledge. The only notable difference is the slightly higher number of liabilities in heavy chains with respect to light chains, which could be explained by the greater diversity and length of the former.

Finally, we contrasted the distribution of specific liabilities across datasets (Fig 3). The distributions are not identical to each other, however, the patterns are broadly similar. A notable exception is the severe extra cysteine (xCys) that is rarer in the therapeutic dataset (1% whereas for other datasets 2.80%-7.59%). Upon examination, we noticed that extra cysteines in therapeutics had an engineered structural role, by bridging the heavy and light chains. It is important to keep in mind that additional Cys residues, especially unpaired, often do not appear in therapeutic antibodies since they can promote aggregation and impede product development by forming intermolecular crosslinks. There are exceptions though, such as palivizumab that has an unpaired buried Cys.

Since we did not note radical differences in numbers or types of liabilities across the datasets, we hypothesized that many of these features could be a result of the most prominent common feature among the datasets, that is germline origin. We calculated how many liabilities one sees on average in each of the heavy human germline subgroups (Table 3). The mean number of liabilities across all datasets is 3.2 (corrected for paired datasets) and is lower than

**Table 1. Antibody Liability Reference.** Liabilities are identified within the IMGT-defined regions in IMGT-numbered sequences.

| Name | Short name/ Tag | Severity | Motif | Description | Citations |
|------|------|------|------|------|------|
| Deamidation (high) | DeAmdH | High | N[GS] in CDRs | Deamidation of Asparagine occurs in the following motifs: NG motif (Asparagine followed by Glycine) and NS motif (Asparagine followed by Serine). Such motifs are known to be associated with deamidation (type of degradation) and can result in reduced "shelf-life". | [7,16] |
| Fragmentation (high) | FragH | High | DP in CDRs | Fragmentation occurs as cleavage at the interface between Aspartate and Proline. It is an example of a common motif that is susceptible to hydrolysis in response to pH. | [17] |
| Isomerization | Isom | High | D[DGHST] in CDRs | Isomerization of Aspartate occurs in the following motifs: DD (Aspartate followed by aspartate), DG motif (Aspartate followed by Glycine), DH (Aspartate followed by Histidine), DS motif (Aspartate followed by Serine) and DT (Aspartate followed by Threonine). Such motifs are known to be connected to isomerization (type of degradation) and can cause a shorter "shelf-life" of antibodies. | [18] |
| Missing Cyst (C) | mCys | High | C not present at 23 or 104 IMGT positions | Missing Cysteine occurs as cysteine absence at IMGT 23 or 104. Certain antibody sequence regions containing unpaired cysteines may result in structural changes, surface charges, or hydrophobicity. | [19] |
| Extra Cys (C) | xCys | High | C present at different position then 23 or 104 IMGT positions | Extra Cysteine occurs as cysteine present at a different position than IMGT 23 or 104. Certain antibody sequence regions containing unpaired cysteines can change an antibody's structure, apparent surface charges, or hydrophobicity. | [20] |
| N-linked glycosylation (NXS/T, X not P) | Ngly | High | N[^P][ST] in variable fragment | N-linked glycosylation occurs as an addition of a sugar molecule. Reduced conformational stability and shorter "shelf-life" of antibody products are connected to asparagine linked glycosylation. Incidence of glycosylation in the CDRs can also directly impair antigen recognition and therefore lead to lower efficacy. | [21] |
| Deamidation (medium) | DeAmdM | Medum | N[AHNT] in CDRs | Occurs in the following motifs: NA (Asparagine followed by Alanine), NH (Asparagine followed by Histidine), NN (Asparagine followed by Aspargine) and NT (Aspargine followed by Threonine). This type of deamidation is less common in comparison to the NG and NS motifs. | [7,16] |
| Hydrolysis | Hydro | Medium | NP in CDRs | Hydrolysis gives rise to the DP motif as a result of the deamidation of Asparagine (N) to Aspartate (D). | [17] |
| Fragmentation (medium) | FragM | Medium | TS in CDRs | Occurs as pH-dependent cleavage at the Threonine—Serine interface. | [17] |
| Trp (W) oxidation | TrpOx | Medium | W in CDRs | Tryptophan oxidation is one of the Post-translational modifications (PTMs). | [22] |
| Met (M) oxidation | MetOx | Medium | M in CDRs | Methionine oxidation occurs in the CDRs. Reduced binding affinity and quicker degradation of the antibody product are linked to oxidation in these particular spots. | [23] |
| Deamidation (low) | DeAmdL | Low | [STK]N in CDRs | Occurs in the following motifs: SN (Serine followed by Aspargine), TN (Threonine followed by Aspargine), and KN (Lysine followed by Aspargine). This type of deamidation is less common than others. | [7,16] |
| Integrin binding | IntBind | Low | GPR\|RGD\|RYD\|LDV\|DGE\|KGD\|NGR in fragment variable | Motifs for following integrin binding: αVβ3 (RGD\|RYD\|KGD\|NGR), α4β1 (LDV), α2β1 (DGE) CD11c/CD18 (GPR). eight human integrins act as RGD receptors: α5β1, α8β1, αVβ1, αVβ3, αVβ5, αVβ6, αVβ8 and αIIbβ3 | [24] |

We used antibody sequences from therapeutics, patents, GenBank, literature, and a large paired next-generation (NGS) sequencing dataset. The therapeutics and patents can be thought of as representatives of the clinical spectrum of sequences [25,26]. GenBank and literature are a mix of antibodies developed for scientific/therapeutic purposes [27,28]. The NGS dataset is a sample of the natural diversity [29]. We extracted unique heavy and light chain sequences from each source and we stratified them by detected organisms (human or non-human closest germline). In the case of the NGS and therapeutics datasets, the heavy and light chains were already paired. All other datasets were unpaired with heavy or light chain sequences.

| | | genbank | | | | | literature | | | | | ngs | patents | | | | | therapeutic | | |
|---|---|---|---|---|---|---|---|---|---|---|---|---|---|---|---|---|---|---|---|---|
| | | H | L | all | human | nonhuman | H | L | all | human | nonhuman | all | H | L | all | human | nonhuman | all | cst | market |
| **sequences** | count | 100.0% | 100.0% | 100.0% | 100.0% | 100.0% | 100.0% | 100.0% | 100.0% | 100.0% | 100.0% | 100.0% | 100.0% | 100.0% | 100.0% | 100.0% | 100.0% | 100.0% | 100.0% | 100.0% |
| | with liabilities | 95.7% | 92.5% | 94.7% | 93.8% | 96.9% | 95.1% | 89.8% | 93.2% | 90.0% | 96.1% | 99.2% | 94.7% | 92.1% | 93.6% | 92.1% | 97.0% | 99.7% | 99.8% | 100.0% |
| | non germline | 84.3% | 74.4% | 81.2% | 83.6% | 75.6% | 84.6% | 66.5% | 78.1% | 77.3% | 78.8% | 93.4% | 87.5% | 74.1% | 82.0% | 83.6% | 78.6% | 96.1% | 97.3% | 96.8% |
| | non therapeutics | 88.0% | 78.9% | 85.2% | 84.5% | 86.6% | 87.5% | 71.9% | 81.9% | 77.1% | 86.2% | 95.7% | 84.2% | 75.5% | 80.6% | 78.5% | 85.5% | 93.5% | 93.1% | 90.3% |
| | not buried | 0.0% | 0.0% | 0.0% | 0.0% | 0.0% | 0.0% | 0.0% | 0.0% | 0.0% | 0.0% | 98.1% | 0.0% | 0.0% | 0.0% | 0.0% | 0.0% | 98.2% | 98.1% | 99.2% |
| | remains filtering | 74.2% | 61.2% | 70.2% | 73.2% | 62.9% | 76.7% | 52.1% | 67.8% | 63.2% | 72.0% | 73.7% | 74.5% | 60.3% | 68.7% | 69.1% | 67.6% | 77.8% | 76.8% | 71.0% |
| **liabilities** | count | 100.0% | 100.0% | 100.0% | 100.0% | 100.0% | 100.0% | 100.0% | 100.0% | 100.0% | 100.0% | 100.0% | 100.0% | 100.0% | 100.0% | 100.0% | 100.0% | 100.0% | 100.0% | 100.0% |
| | non germline | 58.5% | 49.7% | 56.0% | 59.2% | 48.9% | 70.0% | 43.4% | 61.1% | 56.8% | 64.7% | 43.7% | 68.7% | 51.9% | 62.3% | 66.1% | 54.3% | 62.8% | 61.6% | 60.4% |
| | non therapeutics | 64.7% | 65.6% | 64.9% | 67.2% | 59.8% | 72.0% | 59.2% | 67.7% | 65.6% | 69.4% | 61.5% | 63.9% | 60.8% | 62.7% | 63.4% | 61.3% | 54.0% | 52.7% | 47.2% |
| | not buried | 0.0% | 0.0% | 0.0% | 0.0% | 0.0% | 0.0% | 0.0% | 0.0% | 0.0% | 0.0% | 72.1% | 0.0% | 0.0% | 0.0% | 0.0% | 0.0% | 71.8% | 71.2% | 71.6% |
| | remains filtering | 44.9% | 38.0% | 42.9% | 46.0% | 35.9% | 57.0% | 31.4% | 48.4% | 42.2% | 53.7% | 22.6% | 49.0% | 37.6% | 44.6% | 46.8% | 40.2% | 27.9% | 26.3% | 24.1% |

**Fig 1. Per-dataset prevalence of sequence liabilities for five open-source databases: Genbank, literature, NGS, patents, and therapeutics.** Please note that the NGS dataset and therapeutics were paired, so the number of liabilities can not be directly compared to the single-chain datasets. Genbank, patents, and literature datasets contained unpaired heavy and light sequences. In the top portion (sequences) counts are given as a percentage of the total number of sequences in a dataset. In the lower portion (liabilities), the total count of liabilities in the dataset is given. In each case, we show the number of remaining sequences of liabilities or total liabilities after applying individual flags or their combinations.

the mean of 4.12 across germline datasets (Tables 2 and 3). The distribution of liabilities in the germline subgroups is unequal with IGHV1, IGHV3, and IGHV4 having between 2.21 and 3.34 liabilities with IGHV7 as many as 5.85 (Table 3). We checked the germline usage of the subgroups in therapeutics to reveal that the three subgroups with a lower number of liabilities (heavy IGHV1, IGHV3, IGHV4, and kappa IGKV1, IGKV3) are indeed the most commonly

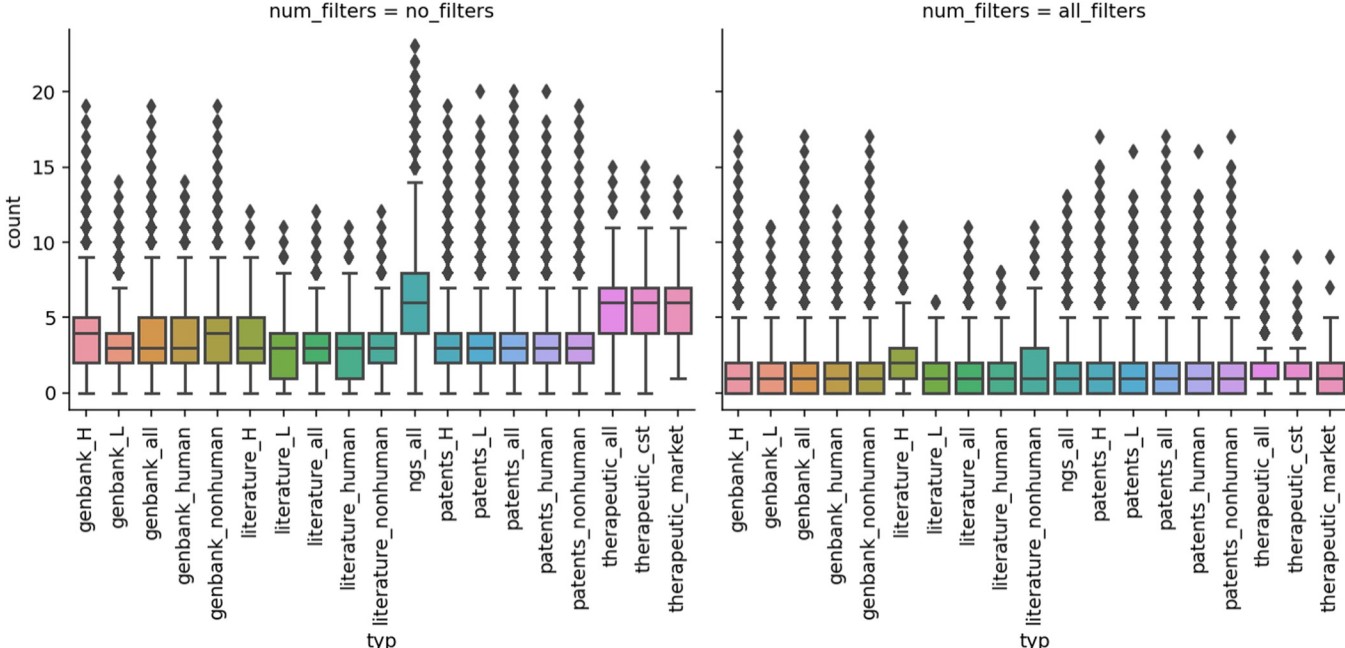

**Fig 2. Per-sequence prevalence of liabilities.** Please note that the NGS dataset and therapeutics were paired, so the number of liabilities can be roughly 2x as many as in single-chain datasets. Unpaired heavy and light sequences were found in the literature, patent, and Genbank datasets. **Left**. Average per-sequence counts of any liability identified in our datasets. **Right**. Average per-sequence counts of any liability identified in our datasets that did not match any of our three flags, therapeutic, germline, or surface (for paired data sets only). Abbreviations after the underscore mean respectively: "H"—heavy chain, "L"- light chain, "all"—all sequences, "human"—only human antibody sequences, "nonhuman"—only non human antibody sequences, "cst"—clinical stage therapeutics, "market"—therapeutics on the market.

**Table 2. The mean number of liabilities per sequence for each dataset in our study.** For most of the datasets, we calculated the mean number of liabilities for unpaired sequences. The NGS and therapeutics subsets offer paired data, which are not directly comparable to single-sequence datasets. Abbreviations after the underscore mean respectively: "H"—heavy chain, "L"- light chain, "all"—all sequences, "human"—only human antibody sequences, "nonhuman"— only non human antibody sequences, "cst"—clinical stage therapeutics, "market"—therapeutics on the market, "std"— standard deviation.

| Dataset | mean | std | median |
|---|---|---|---|
| genbank_H | 3.59 | 1.96 | 4 |
| genbank_L | 3.07 | 1.95 | 3 |
| genbank_all | 3.43 | 1.97 | 3 |
| genbank_human | 3.37 | 2.04 | 3 |
| genbank_nonhuman | 3.57 | 1.79 | 4 |
| NGS_all (both H+L) | 6.55 | 2.96 | 6 |
| literature_H | 3.35 | 1.94 | 3 |
| literature_L | 3.00 | 2.02 | 3 |
| literature_all | 3.23 | 1.98 | 3 |
| literature_human | 3.08 | 2.12 | 3 |
| literature_nonhuman | 3.36 | 1.83 | 3 |
| patents_H | 3.31 | 1.94 | 3 |
| patents_L | 2.95 | 1.92 | 3 |
| patents_all | 3.16 | 1.94 | 3 |
| patents_human | 3.06 | 2.01 | 3 |
| patents_nonhuman | 3.38 | 1.75 | 3 |
| therapeutics_all (both H+L) | 5.85 | 2.62 | 6 |
| therapeutics_cst (both H+L) | 5.91 | 2.61 | 6 |
| therapeutics_market (both H+L) | 6.13 | 2.66 | 6 |

used across therapeutics (Fig 4). Similarly clear distinction is not the case for lambda chains (Fig 4).

We also checked which antibody region harbored the liabilities in the germlines. As expected, most of them are located in the Complementarity-Determining Regions (CDRs), confirming the typical approach of focusing on CDRs for liability detection that can reduce drug efficiency (see Fig 5). It is known that the definition of CDRs and the Vernier zone varies between numbering schemes [30]. This raises a question to what extent our choice of IMGT numbering for CDR definition affects liabilities being missed in the Vernier zone. Fig 5 shows that there are indeed liabilities in the Vernier zones, namely CDR-H3 to FW-H3, FW-L1-CDR-L1, and CDR-L2-FW-L2. In each case liability within the boundary meant one residue in the CDR and the other in the framework. Otherwise, the number of liabilities beyond CDRs and such one-residue boundaries is very low. Therefore, employing the IMGT definition accounting for +1 residue around the CDRs accounts for most of the liabilities.

Altogether, our results demonstrate that identifying liabilities by simple sequence motifs reveals a multitude of potential issues with many of these, likely to be of germline origin. There appears to be no dataset-specific pattern. The large proportion of universally present liability motifs confirms that the majority of them are likely to be false positives if the predictions are made using the amino acid sequences of the antibody light and heavy chains alone. Despite this limitation, sequence-based physicochemical liability predictions are a useful tool at the earliest stages of biological drug discovery where a large number of potential hits need to be prioritized for the initial experimental characterization. Such predictions could be further refined by considering the importance of the chemical degradation liability and its location in the antibody sequence. Furthermore, performing antibody structure predictions and

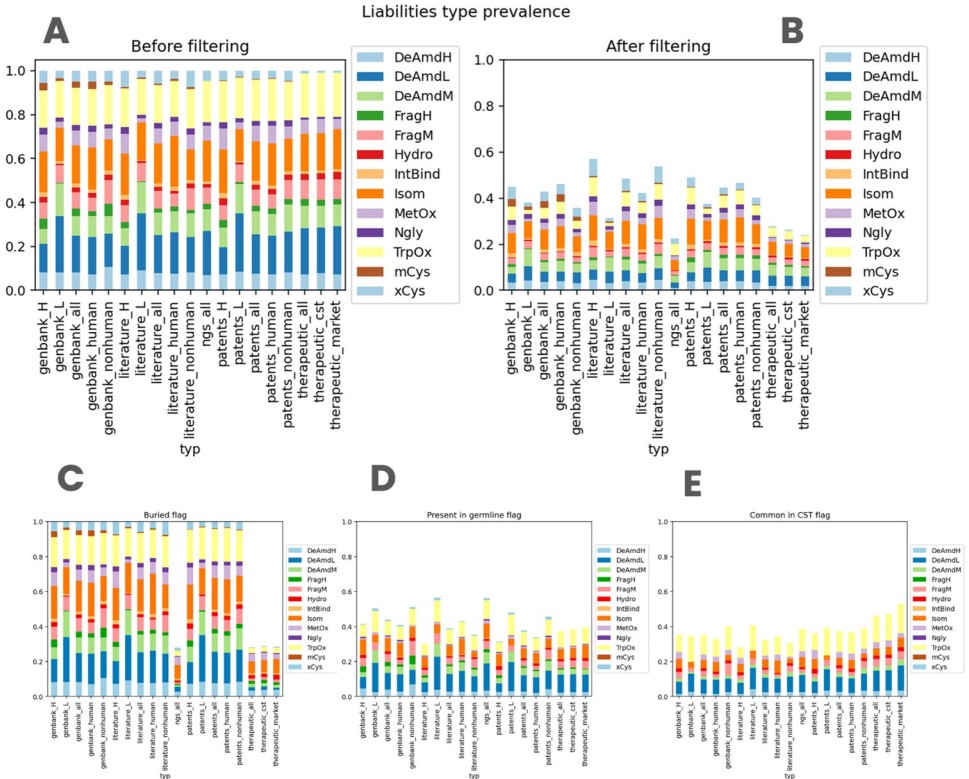

**Fig 3. Prevalence of specific sequence-based liabilities across our datasets.** Abbreviations after the underscore mean respectively: "H"—heavy chain, "L"- light chain, "all"—all sequences, "human"—only human antibody sequences, "nonhuman"—only non human antibody sequences, "cst"—clinical stage therapeutics, "market"—therapeutics on the market. **A**. Liability distribution per dataset without applying any flags. **B**. Liability distribution per dataset after applying all three flags at once. **C**. Liability distribution per dataset after applying the 'buried' flag (Note that only the rightmost datasets with paired heavy/light chains are affected). **D**. Liability distribution per dataset after applying the 'germline' flag. **E**. Liability distribution per dataset after applying the 'therapeutic' flag.

estimating solvent accessibility of the chemical degradation motifs can also help reduce the rate of false positives in the sequence-based liability predictions.

## Leveraging therapeutic, germline, and structural information to reduce the number of potential false positives

The high prevalence of sequence liability motifs across all databases, including therapeutics, suggests that a considerable proportion of these could be false positives. To focus on de-risking

**Table 3. Prevalence of liabilities in human germline immunoglobulin heavy subgroups.** For each set of germline sequences associated with a particular subgroup, we counted the number of liabilities it harbored.

| subgroup | Mean liabilities | Std liabilities | Median liabilities |
|---|---|---|---|
| IGHV4 | 2.21 | 1.07 | 2 |
| IGHV3 | 2.60 | 1.22 | 2 |
| IGHV1 | 3.34 | 1.48 | 3 |
| IGHV6 | 4.36 | 0.49 | 4 |
| IGHV5 | 5.25 | 1.10 | 5 |
| IGHV2 | 5.36 | 1.07 | 5 |
| IGHV7 | 5.85 | 0.71 | 6 |

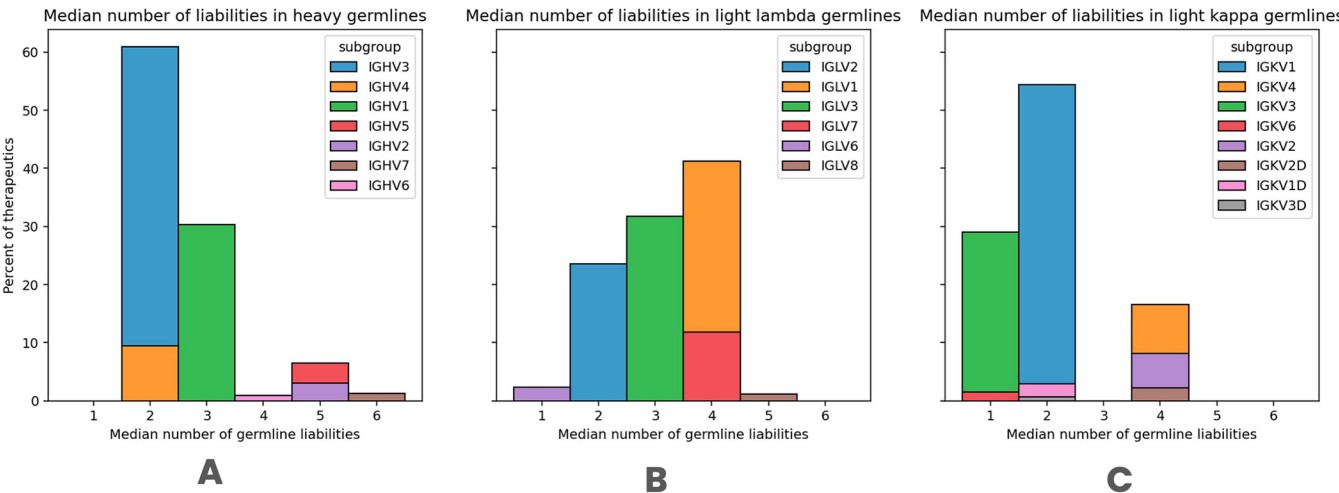

**Fig 4. The median number of germline liabilities in therapeutics.** We counted the number of detected liabilities in each therapeutic sequence. These were further stratified by detected germlines. We show the number of liabilities for A) Heavy chains B) Light lambda chains and C) Light Kappa chains.

molecules one needs to constrain the set of examined liabilities only to those that have a higher probability of detrimental chemical modifications, in the experimental characterization studies. Here, we employed three intuitive liability flags that are used within the industry for filtering the sets of liabilities. These are *germline*, *surface*, and *therapeutic* flags (Fig 6).

The germline flag examines whether a given sequence motif is present in the germline gene sequence at the same position. If so, it is plausible to assume that it may not significantly hinder the function of naturally occurring antibodies derived from it, even though such liabilities may still contribute to drug product heterogeneity, which is a major concern during drug development. The surface flag employs structural information to examine whether a given sequence motif is exposed to solvent or not (fully or partially buried). Water is known to drive

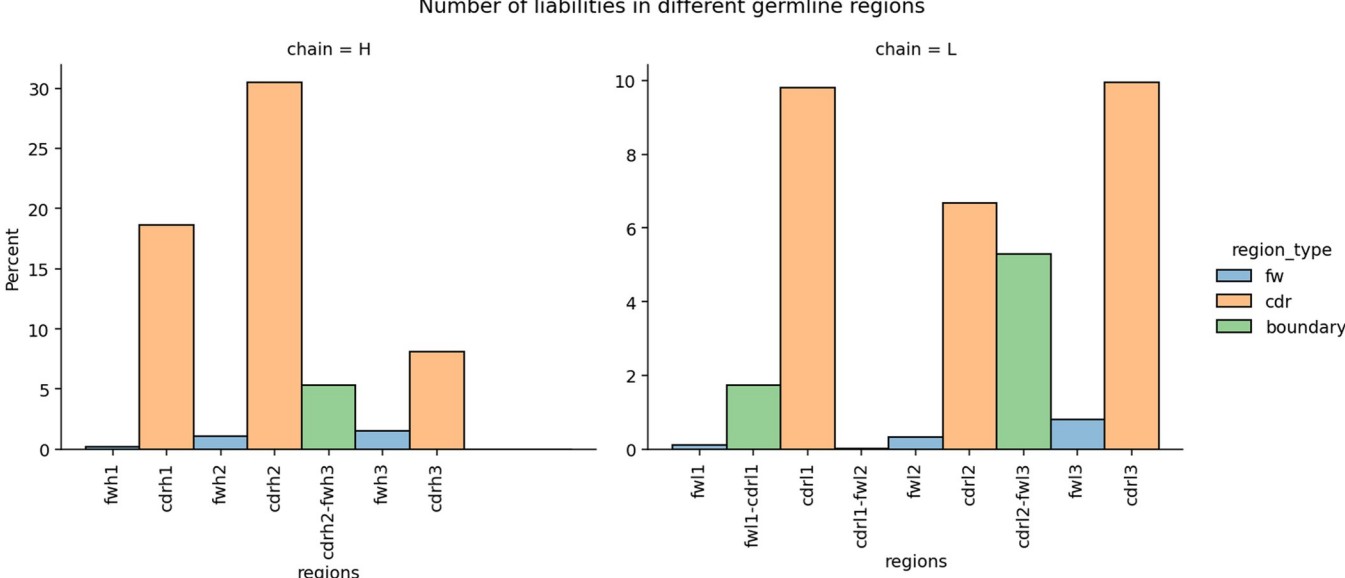

**Fig 5. The number of liabilities in different germline regions.** The boundary region indicates liabilities that are two-amino acids long where the first was found in the framework and the second in the CDR region. **Left.** The number of liabilities in a heavy chain. **Right.** The number of liabilities in a light chain.

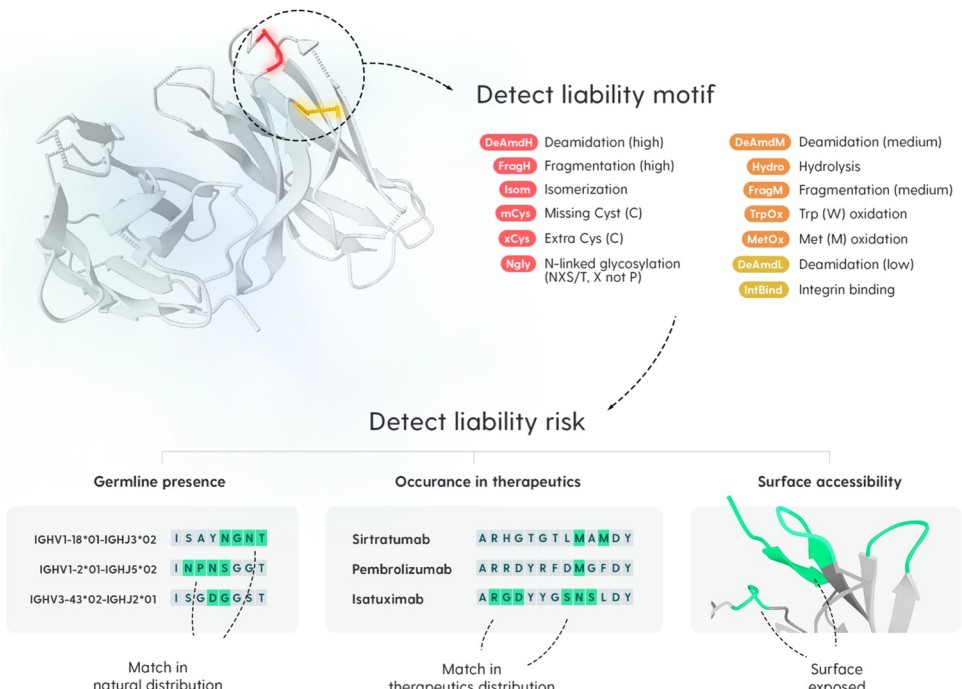

**Fig 6. Liability detection and low-risk flags.** We use our Antibody Liability Reference to detect sequence liability motifs in query antibody sequence. We cataloged 70 liability motifs with different severity levels (the colors mean: red—high severity, orange—medium severity, and yellow—low severity). Afterwards, three flags are applied, which are designed to convey an intuitive association with lower risk despite motif presence. Germline presence flag is set to true if the liability is also found in a germline reference for the given sequence. Therapeutic flag indicates how common a motif is in marketed therapeutics. Surface accessibility flag indicates whether the motif is buried, partially buried, or exposed according to a three-dimensional model.

the chemical degradation of proteins [31] and therefore access of a liability motif to water can increase its susceptibility to chemical degradation. Consistently, solvent accessibility has been shown to be a good predictor thereof [13]. This flag requires a reasonably accurate structural model to be produced, which can be achieved by one of many structural modeling tools available these days [32,33]. Prevalence in therapeutics is a flag introduced by Jacobitz et al. [9]. Here, the authors propose that if a sequence motif occurs at a specific position in marketed antibodies, it should potentially pose a lower risk because the therapeutic made it through the clinic and manufacturing process despite its presence due to successful formulation development. However, one should keep in mind that such liabilities may still contribute towards the drug product heterogeneity and therefore affect batch-to-batch comparability studies. Sequence-structural context of a physicochemical degradation motif specific to a particular drug candidate may still lead to it being labeled as a critical quality attribute (CQA) for the drug candidate, even though it is already found in the approved biotherapeutics. Approval of novel biotherapeutic drugs is often granted on a case-to-case basis after comprehensive reviews of multi-factorial risk versus benefits analyses performed by the regulatory agencies and regulatory expectations continue to evolve. Therefore, care should be taken when considering the presence of a given chemical liability motif in approved biotherapeutic(s) as benign.

For any liability flag to be useful, it must considerably reduce the number of liabilities one needs to examine via experiments. For this reason, we benchmarked the ability of different flags, individually and in combination, to constrain the number of liabilities one would have to investigate.

Employing the human germline flag reduces the number of human sequences with any liabilities by ca. 5–20 percentage points across the databases (Fig 1). The reduction is much more pronounced considering the total number of liabilities removed. Between 40–60% of all liabilities across our databases can be marked with the germline flag. This indicates that a significant portion of the sequence liability motifs present across antibodies occurs naturally.

Using the therapeutic flag reduces the number of sequences with any liabilities by a similar amount to germline flags (5–20%). This comes with the caveat for numbers presented for therapeutics in Fig 1, as the 'training' set for the flag is included in this dataset. In terms of the total number of liabilities removed, the therapeutic flag appears to reduce a lower number than germline, ca. 40 percentage points. Despite the relatively small number of marketed therapeutics, motifs present in these appear prevalent in other datasets.

The therapeutic and germline flags are sequence-based annotations and thus can be applied to single sequences in the absence of a cognate heavy/light chain pairing. Application of the surface flag requires the presence of the cognate heavy/light chain pairs, otherwise, residues on the heavy/light interface would be incorrectly assigned as exposed. Therefore we only applied the surface flag to datasets where we had the paired data, namely the therapeutic dataset and the natural, 1.3 million paired sequences from Jaffe et al. study [29]. For structural modeling we employed ABodyBuilder2 as it offers good quality and is freely available, facilitating the reproduction of our results [34].

Employing the surface flag, the total number of sequences without any liabilities decreases by around 2 percentage points for natural sequences and by approximately 2 percentage points for the therapeutic ones when the surface flag is used. When individual liabilities are taken into account, the effect is stronger; for both the natural and therapeutic datasets, the decline is in the range of 30 percentage points (Fig 1). Therefore, the surface flag has a non-trivial capacity to flag liabilities, however, it is less likely to leave a molecule without any liabilities. On the same datasets, the surface flag reduces fewer liabilities overall than germline or therapeutic flags, at a much higher computational overhead because of the higher cost of creating the structural model (comparing the results on the paired datasets for consistency).

We established that each of the flags has the ability to mark a non-trivial number of potentially false-positive liabilities. If the flags mark the different types of residues, then employing them simultaneously could bring additional benefits. Therefore we checked the ability of the combinations of the flags to study the extent to which they overlap in reducing the number of residues to examine. We checked the combination of therapeutic and germline as it was not obvious whether frequent positions in therapeutics were simply a recapitulation of the germline frequency. We then use all three flags at once to check the benefit of enriching the sequence-derived flags (therapeutics and germline) with structural information.

Employing both the germline and therapeutic flags reduced the number of sequences with any liabilities by ca 10–20 percentage points with respect to using any of the flags individually. The drop was more dramatic considering individual liabilities where combining the flags can reduce the total number of positions to examine to only 30% (Fig 1). Therefore, germline and therapeutic flags hold sufficiently dissimilar sequence information, so using them both appears to provide more potential benefit than using any of them individually.

Introducing structural information in the form of the surface flag to the combination of therapeutic and germline flags further reduces the total number of sequences with any liabilities than using the combination of the sequence flags alone. The number of natural sequences whose liabilities do not fit our flags is 73.7% (dropped from 1.332.050 to 981.658) whereas for therapeutics it is 77.8% (from 618 to 481). The combination of the three flags reduces the total number of liabilities one would have to examine to just 22.6% (1.973.043 out of 8.731.105) in natural sequences and to 27.9% in all therapeutics (1010 out of 3615). Overlap among different

flags for both paired and unpaired datasets is presented in S2 Fig, demonstrating that all flags appear to complement each other. Therefore, structural information complements sequence information, providing the largest reduction in the total number of pertinent liabilities to be examined.

On the individual sequence level, all the flags bring the median of the total number of liabilities from 3–4 to 1–2 (Fig 2). This drop appears to be consistent across datasets, flags employed, and their combinations.

The number of sequences with liabilities (Fig 1, row "sequences remaining filtering" for literature heavy chain, therapeutics all and Clinical Stage Therapeutics (CST) are slightly higher than for others (they have more than 76% whereas all others are below 74%). This is consistent with box plots in Fig 2 as those three datasets have higher 1st quartile values (value of 1 whereas all others have 0). The flags employed do not appear to favor a single liability type over another as shown in Fig 3. After applying all the flags, the total number of liabilities drops by 60% in most cases, however, the distribution of the liability types remains broadly consistent.

Our results demonstrate that our flags identify close to 60% of all liabilities as innocuous, independent of dataset and liability type. In practical terms, it reduces the number of identified liabilities to examine from ca. 3–4 per sequence to 1–2 per sequence, that are presumed to be higher risk.

## Benchmarking flagged liabilities on their ability to predict positions less susceptible to chemical modifications

We established that employing our three liability flags filters out a significant proportion of liabilities and leaves only the ones that are presumed to be higher risk. To check whether such a distinction is indeed the case, we benchmarked the ability of LAP flags to remove liabilities with lower susceptibility to chemical modifications on an experimental dataset. For this purpose, we employed Lu et al. Isomerization/Deamidation dataset [7], which consists of clinical stage biotherapeutics with measured deamidation and isomerization events under pH stress.

We employed data for three measurements: isomerization at low pH, deamidation at high pH, and deamidation at low pH. Each event was associated with a list of therapeutics and their modifications. We removed the therapeutics that were employed in the creation of our therapeutic flag. Likewise, certain sequence motifs are not part of our liabilities reference, and these were left out. Table 4 gives a breakdown of the original number of therapeutics and modifications and the ones we employed in benchmarking the flags.

For each of the sequence liabilities, we note whether LAP would flag it as being lower risk. In Fig 7 we show the percent of modification of the sequence motifs without applying LAP,

**Table 4. The number of therapeutics and liabilities used to assess the predictive power of the LAP flags on the Lu et al.** Isomerization/Deamidation dataset [7]. Since the therapeutic flag employs some of the therapeutics that were screened in Lu et al. Isomerization/Deamidation dataset [7], we had to check LAP performance with and without these sequences. The numbers in the table below show the number of therapeutics and the associated measured liability data points with or without the therapeutics that were employed in the construction of the LAP therapeutic flag. "LAP" stands for Liability Antibody Profiler and "CST" stands for Clinical Stage Therapeutics.

| Liability | LAP CSTs removed | | | | Keeping all CSTs | | | |
|---|---|---|---|---|---|---|---|---|
| | Therapeutics to start with | Therapeutics used | All liabilities | Used liabilities | Therapeutics to start with | Therapeutics used | All liabilities | Used liabilities |
| Deamidation high pH | 33 | 17 | 39 | 14 | 33 | 33 | 39 | 28 |
| Deamidation low pH | 18 | 12 | 21 | 11 | 18 | 18 | 21 | 16 |
| Isomerization low pH | 28 | 17 | 31 | 15 | 28 | 28 | 31 | 23 |

## LAP CSTs removed

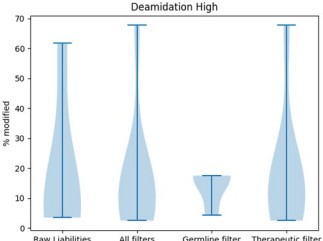 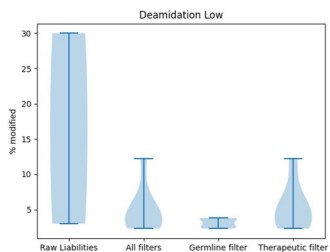 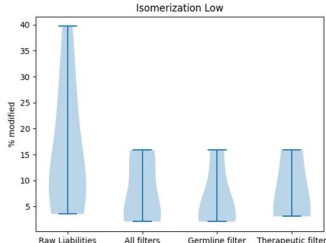

## Keeping all CSTs

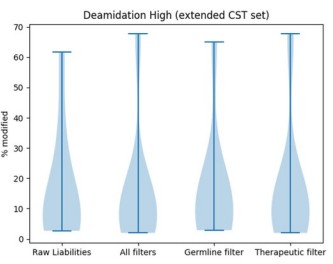 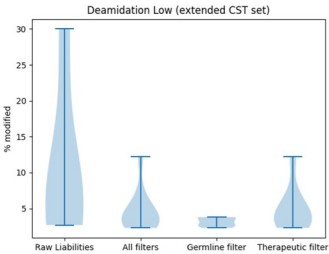 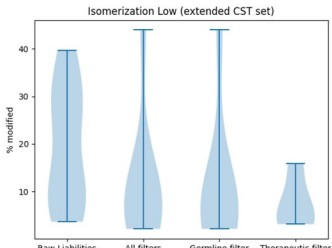

**Fig 7. Benchmarking the predictive ability of the Liability Antibody Profiler (LAP) flags to filter innocuous liabilities.** The y axis shows the distribution of the percentage of each liability undergoing modification. In most cases, liabilities are associated with one or more LAP flags. We only show the germline and therapeutic flag individual distributions as no liabilities in the Lu et al. Isomerization/ Deamidation dataset and Liability dataset were detected to be buried. "CST" stands for Clinical Stage Therapeutics.

applying all LAP flags, and applying one flag at a time. Note that we have not detected any buried residues in this dataset so the surface flag does not apply here.

Fig 7 shows that the sequence liabilities that were flagged have lower susceptibility to chemical modification. This is more pronounced for the deamidation at low pH and isomerization at low pH cases, which generally have lower rates of modification, as expected. We do not see an effect in the deamidation at high pH. Presence in the germline appears to be the strongest individual flag that consistently indicates lower frequency modifications even in the high severity deamidation. Therapeutic-derived modifications do not show a strong effect in the deamidation at high pH. Here, there appears to be no pronounced benefit of combining both germline and therapeutic flags. For comparison, we have plotted the same graphs but without removing the therapeutics that were part of the therapeutic flag training set, which does not change the results radically. For each flag combination (all flags, germline, therapeutic) we compared the set of frequencies with or without CSTs using two-sample t-tests. The lowest p-value we obtained was 0.51 in Isomerization Low case, indicating that though the distributions could appear to have seemingly different shapes their means are not statistically different.

We did not have the opportunity to benchmark the surface flag on the Lu et al. Isomerization/Demidation dataset [7] since there were no buried motifs there. For this reason, we also employed the Yang et al. Oxidation dataset [35]. Here, the authors subjected 121 clinical stage antibodies to forced oxidation (0.1% H2O2 for 24 hours). Authors benchmarked a machine learning surface exposure predictor on its ability to link surface exposure of methionine (commonly associated with oxidation event) with the oxidation of the molecule.

**Table 5. Benchmarking the performance of prediction of methionine oxidation.** For each subtable (oxidation/no oxidation versus with flag/no flag) we performed the Fisher's exact test. The result for the germline flag is deemed not to be statistically significant as per Fisher's exact test.

| | Germline Flag (not statistically significant) | | Surface Flag | | Therapeutic Flag (not applied to those in training set) | | All flags (therapeutic flag not applied to those in training set) | |
|---|---|---|---|---|---|---|---|---|
| | With flag | No Flag | With flag | No Flag | With flag | No Flag | With flag | No Flag |
| Oxidation | 5 | 26 | 10 | 21 | 5 | 26 | 12 | 19 |
| No Oxidation | 4 | 17 | 17 | 4 | 10 | 11 | 19 | 2 |

Here we benchmarked our three flags on their ability to also link the oxidation event with methionines we detected, with results given in Table 5. We only counted the therapeutics where we annotated methionines according to our Antibody Liability Reference. Predicting no oxidation event where there is no methionine would otherwise count such cases as true negatives which would be reflected as the better predictive power of our method whereas no prediction was performed. We tested the ability of each of the three flags, in turn, to sort the oxidation/no oxidation molecules by virtue of methionine flagging and their combination. Flagging a methionine should be considered as calling lower risk of oxidation. Each of the resulting 2x2 tables was subjected to Fisher's exact test, with only the germline flag (which performed quite poorly), failing the significance test. The surface flag did the best job of correctly annotating most of the methionines in molecules associated with oxidation, with roughly 50% accuracy on those that did oxidize. Combining all the flags appears to maintain the best annotation of methionines with molecules that do not oxidize as flagged and those that oxidize as not flagged.

Altogether, our results show that our simple flags, though not streamlined nor trained on any particular experimental dataset, offer reasonable performance in predicting a range of positions and their susceptibility to modifications.

### Online tool for liability annotation

To facilitate access to our LAP protocol we built a web application available at https://lap.naturalantibody.com. Users can submit individual sequences of heavy and light chains, which will be IMGT-numbered, CDRs annotated according to the IMGT definition. All the detected liabilities are annotated on the sequence. If a single heavy or light sequence is submitted, only the germline and therapeutic flags will be applied. If both heavy and light chains are supplied, the structural model is created using ABodyBuilder2 [34], and the surface flags are calculated. The liabilities contextualized to the structure can then be inspected in a 3D molecular viewer.

### Discussion

Biotherapeutics need to be not only functional but also developable (safe, effective, and manufacturable) for their successful clinical applications. Therefore, delivering a biologic drug product to the market requires multi-faceted optimization including therapeutic concept, pharmacology, and developability (safety, efficacy, and manufacturability). Annotation of sequence liabilities is a widespread practice across early-stage antibody discovery and engineering. Entire phage libraries can be designed as liability-free [36] and much effort is exerted to engineer these out of individual sequences during the lead optimization stages [11,37,38].

Despite the centrality of physicochemical degradation challenges towards maintaining molecular integrity of the drug substance during manufacturing, storage, and shipping, delivery, and administration, there does not exist a 'golden reference' for sequence motifs that underpin chemical liabilities and their severities for biotherapeutics. Our compilation of these,

though aimed at consolidating knowledge from fragmented sources, is far from ideal. Different manufacturers have distinct qualifications for the severity of different liabilities. Commonalities exist, such as unpaired cysteines that are considered universally detrimental. The list of motifs employed here is not exhaustive as motifs outside of this compilation were also noted to undergo chemical modifications [7]. Moreover, this list is likely to continually grow as newer technologies for bioanalytical characterization of drug product heterogeneity of biotherapeutics emerge either independently or in response to regulatory guidelines.

Based on our Antibody Liability Reference, our analysis of liability motif frequency from all major sources is in line with observations coming from phage libraries noting that hardly any sequence is free of sequence liability motifs [39]. Therefore identification of chemical liabilities purely by sequence occurrence is an oversimplification that can lead to many false positives. Engineering sequences without any liabilities is challenging and might not account for previously unknown chemical degradation motifs that are not part of any reference [7]. However, one should note that Teixeira and coworkers [36] have recently created antibody display libraries with maximal developability and liability free CDRs.

The other side of the extreme of removing all sequences with liability motifs is predicting which ones one can tolerate despite their presence [9]. For example, the location of a chemical liability motif in the protein 3D structure, its interactions with the neighboring residues, solvent, and characteristics of the drug product formulation can significantly lower its ability to undergo chemical modification during the shelf-life of the drug product. Deciding on which motifs can be tolerated and which ones must be removed during the lead optimization stages of the lead candidates can be facilitated by the machine learning based predictions [13,14,35]. However, the machine learning methods are currently focused on individual modifications rather than covering their entire spectrum. Insofar as they provided an excellent proof of concept, however, the full application of machine learning in liability detection relies on large proteomic datasets that cover a comprehensive spectrum of chemical modifications.

Though larger experimentally derived chemical degradation datasets for therapeutic antibodies are welcome, one needs to be aware of their caveats. It is challenging for mass spectrometry-based methods to pinpoint individual liability motifs that are in fact degrading, especially when two motifs occur close together in sequence. For instance, double-motif NGNT is often found in the light chain CDRs of the antibodies but it is hard for mass spectrometry to reliably decipher which asparagine is degrading. Furthermore, forced degradation data might not translate into real-time chemical degradation data from the stability studies. Formulation composition can also significantly alter the physicochemical degradation rates for an antibody. For instance, citrate-free formulation of Humira is superior to the original one [40]. All such considerations need to be taken into account when interpreting the experimental dataset benchmarking of this or any other liability study. Even a strong predictor will likely be tied to specific experimental conditions on which it was trained.

Here, we benchmarked a solution offering a balance between over-predictive annotation of all sequence motifs and machine learning methods operating on narrow available datasets. Our three flags, with intuitive interpretations of structural (surface), natural (germline), and therapeutic (marketed occurrence) were shown to considerably reduce the number of liabilities to be examined by close to 60% when all of them are considered at once.

We envisage further modifications to the liability scoring schemes, employing novel techniques such as language models trained on plentiful NGS-derived data of natural provenance [26,41,42]. However, at this stage we avoided such techniques and data in favor of simple-to-interpret flags. With regard to natural antibody data, we employed the simpler germline flag as we questioned whether any NGS-observed naturally occurring mutations exhibit a favorable liability profile, suggesting that this might vary based on observed frequencies. Certain motifs

may be less frequent due to the potential for heterogeneous binding caused by chemical modifications, but this phenomenon would need to be quantified experimentally. Considering that germlines represent a uniform element in the diverse and individualized array of immunoglobulins, we propose that the acceptance of certain components in the germline indicates a substantial biological rationale.

Another modification that we envisage is model-driven, especially making use of the language models [43]. The annotations that we offer are devoid of the larger sequence context which language models capture. However to make full use of such models, one would require plentiful experimental annotations. Whilst lacking in plentiful experimental data, our introduced flags make the best use of the datasets available at the moment, not for training but benchmarking purposes.

We hope that our study of liabilities across diverse datasets and tools we develop will facilitate this one facet of engineering novel biologics.

## Materials and methods

### Public datasets

We employed five public generalistic datasets, patents, GenBank, literature, therapeutics, and NGS. Literature, Patent, and Genbank datasets consisted of unpaired heavy and light sequences.

Similarly to [25], patent sequences were imported from WIPO, USPTO, DDBJ, and PSIPS in heterogeneous formats. WIPO File Transfer Protocol (FTP) contained multiple data formats of which only sequence listings were selected.

Amino acid sequences of length below 40 or above 1000 were discarded.

Furthermore, sequences were discarded based on any of the following conditions: containing codons different than 20 standard aa, containing ambiguous codons, and containing the '!' sign.

Sequences were numbered using ANARCI, and sequences meeting any of the following criteria were removed: no germline could be assigned to the sequence, no heavy or light chain was found in input sequence, two chains were of the same type (e.g. heavy+heavy), not all CDR's were present in the numbered sequence. This procedure rendered a patent dataset which consisted of 174.207 heavy and 120.851 unique light chain sequences (Table 6).

Similarly to what was described previously [44], GenBank files were imported from NCBI FTP server, then coding region annotations were extracted to obtain AA sequences. Sequences obtained this way followed the same filtering and numbering pipeline as described above, to produce 129.434 heavy and 58.599 unique light chain sequences (Table 6). The GenBank and Patent sequences were further stratified by the human and non-human gene annotations.

Literature dataset is a heterogeneous collection of sequences manually mined from literature at NaturalAntibody and consists of a total of 4,757 antibodies (Table 6). For this dataset, additionally, IgBLAST was used to obtain AA sequences from nucleotides.

NGS and therapeutics datasets consisted of paired heavy and light data. The NGS dataset consisted of 1,332,050 paired human antibody sequences from a single study [29] (Table 6). Therapeutic antibody sequences are collected from the WHO INN lists as described previously

**Table 6. Number of unique sequences per dataset.** Unique sequences were calculated on the basis of the uniqueness of their variable region sequences for single-chain datasets and the concatenated chains for the paired datasets.

|  | Patents | GenBank | Literature | NGS | Therapeutics all |
|---|---|---|---|---|---|
| Heavy chains | 174.207 | 129.434 | 3.022 | 1.325.126 | 608 |
| Light chains | 120.851 | 58.599 | 1.707 | 498.646 | 590 |

[26] that are not discontinued. Clinical Stage Therapeutics (CST) are further defined as a subset being post phase I and consequently Market Stage Therapeutics (MST) are those being approved. In all cases, tetraspecific molecules are considered as two independent bispecifics, and bispecifics are treated as two independent antibodies. Therefore we had three therapeutic datasets, all, CST, and marketed with 618, 479, and 124 molecules respectively.

### Experimental liabilities datasets

We extracted therapeutic sequences associated with three liabilities, deamidation at low pH, deamidation at high pH, and isomerization at low pH from a previous study [7], termed the Lu et al. Isomerization/Deamidation dataset. Each liability motif was associated with an IMGT position and percentage occurrence of a given chemical modification. The therapeutics were further stratified by the ones that occurred in the dataset used for the therapeutic flag with the numbers given in Table 4.

Yang et al. Oxidation dataset was extracted from a study of oxidation effects on 121 therapeutics [35]. We have removed sonepcizumab from this dataset as we did not have it in our WHO-INN therapeutic database. Authors measured oxidation of the Fab portion as well as the constant portion. Since our analysis only encompasses the variable portion, we employed oxidation annotations that authors labeled as 'Fd'.

### Identification of Liabilities—Antibody Liability Reference

To the best of our knowledge, there is no 'golden reference' for sequence liabilities. Rather, different organizations have distinct approaches to annotating these, though the changes are not radical (motifs and severities overlap to a large extent). For this reason, we have created our consensus Antibody Liability Reference liability by sourcing epistemological knowledge from colleagues and co-authors. The resulting Table 1, aimed at keeping the commonalities between various definitions.

The motifs presented in Table 1. are based on two types of sources. The first are internal databases of co-authors' liabilities motifs that are used in their organizations. The other one is a compilation from scientific literature [7,9,13,36,45,46].

We cataloged a total of 70 motifs, of which we used 30 for further analysis that were repeated between sources or had strong scientific evidence for their characteristics. In case of any ambiguity, we consulted with co-authors to decide on the classification of a given motif.

### Liability annotations and flags

All the sequences are IMGT-numbered using ANARCI [47] employing the VDJBase germline sets [48]. We used a single version of the numbering software throughout all database and LAP annotations, as differences in annotation versions and underlying germline datasets can result in inconsistencies in resulting numberings [49]. The numberings delineate the IMGT CDR regions that are employed for liability annotations. We are aware that different CDR definitions can result in different liability annotations and the IMGT scheme was used for consistency. Sequence liabilities are annotated using Antibody Liability Reference motifs in Table 1 if they occur within any of the regions associated with a given motif. Each of the motifs is then subjected to one of the three flags, germline, therapeutic, or structural.

### Germline

If a given motif is found in a human germline sequence at the same IMGT position, it is flagged as 'matched'. Such motifs are conjectured to have natural provenance and should pose

a smaller risk within the context of a living organism. Please note that this flag only considers the V and J regions because of the lower fidelity of the D region identification and the reliability of the subsequent alignment. Thus, most of the CDR-H3 is not captured for consideration by the germline flag. However, CDRs-L1, L2, L3, H1, and H2 are included in the germline flag, as also the framework regions.

### Therapeutic

If a given motif is found in more than 5% of marketed therapeutics at the same IMGT position, it is flagged as 'matched'. The protocol was adapted from previous work by Amgen [9]. Such motifs are believed to pose less risk since molecules with the same motif, though perhaps different sequence contexts, passed all the clinical hurdles.

### Surface

The structural model is created employing ABodyBuilder2 [34] because it is state of the art and has a permissive license allowing for distribution via our website. Employing the 3D structure, We detect whether a given position is more than 7.5% relative solvent accessible surface area (SASA), in which case a position is labeled as exposed, otherwise, it is buried. If a motif has multiple positions, one of which is exposed and the other buried, the entire motif is flagged as 'partial'. This protocol was loosely adapted from a previous study on the effects of surface accessibility on chemical modification propensity [13].

## Supporting information

**S1 Fig. Statistical comparison of the number of liabilities in datasets using chi-squared test (chi2) test with Bonferroni correction applied.** Chi2 test was applied to test the difference in number of liabilities per sequence between any two datasets. Because there were multiple tests, Bonferroni correction was applied. Abbreviations after the underscore mean respectively: "H"—heavy chain, "L"- light chain, "all"—all sequences,"human"—only human antibody sequences, "nonhuman"—only non humanantibody sequences, "cst"—clinical stage therapeutics, "market" -therapeutics on the market.
(TIF)

**S2 Fig. Overlap of different flags for NGS dataset (left) and all therapeutics (right).**
(TIF)

## Author Contributions

**Conceptualization:** Tadeusz Satława, Mateusz Tarkowski, Sonia Wróbel, Paweł Dudzic, Tomasz Klaus, Marek Orłowski, Anna Kostyn, Sandeep Kumar, Andrew Buchanan, Konrad Krawczyk.

**Data curation:** Tadeusz Satława, Mateusz Tarkowski, Paweł Dudzic.

**Formal analysis:** Tadeusz Satława, Mateusz Tarkowski, Sonia Wróbel, Paweł Dudzic, Tomasz Gawłowski, Tomasz Klaus, Marek Orłowski, Anna Kostyn, Sandeep Kumar, Andrew Buchanan.

**Investigation:** Tadeusz Satława, Sonia Wróbel, Paweł Dudzic, Tomasz Gawłowski, Marek Orłowski, Sandeep Kumar, Andrew Buchanan, Konrad Krawczyk.

**Methodology:** Tadeusz Satława, Sonia Wróbel, Paweł Dudzic, Tomasz Klaus, Anna Kostyn, Konrad Krawczyk.

**Project administration:** Konrad Krawczyk.

**Resources:** Konrad Krawczyk.

**Software:** Mateusz Tarkowski.

**Supervision:** Konrad Krawczyk.

**Visualization:** Tomasz Gawłowski.

**Writing – original draft:** Tadeusz Satława, Mateusz Tarkowski, Sonia Wróbel, Paweł Dudzic, Tomasz Gawłowski, Tomasz Klaus, Marek Orłowski, Anna Kostyn, Sandeep Kumar, Andrew Buchanan.

**Writing – review & editing:** Tadeusz Satława, Mateusz Tarkowski, Sonia Wróbel, Paweł Dudzic, Tomasz Gawłowski, Tomasz Klaus, Marek Orłowski, Anna Kostyn, Sandeep Kumar, Andrew Buchanan, Konrad Krawczyk.

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
