## [Decision Letter · Decision Letter 0]

1 Jan 2024

Dear Dr Krawczyk,

Thank you very much for submitting your manuscript "LAP: Liability Antibody Profiler by sequence & structural mapping of natural and therapeutic antibodies" for consideration at PLOS Computational Biology.

As with all papers reviewed by the journal, your manuscript was reviewed by members of the editorial board and by several independent reviewers. In light of the reviews (below this email), we would like to invite the resubmission of a significantly-revised version that takes into account the reviewers' comments.

We cannot make any decision about publication until we have seen the revised manuscript and your response to the reviewers' comments. Your revised manuscript is also likely to be sent to reviewers for further evaluation.

Sincerely,

Nir Ben-Tal

Section Editor

PLOS Computational Biology

Reviewer's Responses to Questions

**Comments to the Authors:**

Reviewer #1: The authors report a comprehensive computational method for identifying liability prone sequence motifs in antibodies. The manuscript is well-written and does an excellent job of bringing together data/methods from disparate sources to explore how sequence can be used to identify liabilities. I recommend publishing this manuscript after the authors address the following minor comments:

1. It isn't clear how the authors arrived at the motifs defined in Table 1. Does this information come from conversations and/or unpublished work or are there published papers that the researchers can cite to clarify this? The value of this method is dependent on the validity of these motifs, so it would be useful to understand where they come from.

2. The authors compare and contrast the distributions of the liabilities across the datasets and then subsequently present a flagging methodology for removing benign liabilities. It would make more sense to show the distributions of liabilities in each dataset after the flagging procedure is applied.

3. On line 239, the authors point to concerns related to CDR numbering schemes. Do the methods used to arrive at Table 1 use the same numbering scheme (IMGT)? Additionally, it would be useful to provide a reference for this, for example:

Dondelinger et al. Understanding the Significance and Implications of Antibody Numbering and Antigen-Binding Surface/Residue Definition, Front. Immunol., 9(2278)1-15, 2018

4. On line 294, the authors state "Aqueous solvents such as water are known to drive the chemical degradation of proteins." Neutral water by itself without co-solutes does not generally drive chemical degradation of proteins. Also, aqueous by definition refers exclusively to water and no other solvents. I believe the intention of the authors was to point to the fact that aqueous (water containing) solutions (not solvents) can lead to chemical degradation, so I think this should be rephrased to make it more clear.

5. The phrasing on line 235 is awkward, consider rephrasing to "which antibody region the liabilities in the germlines were in".

Reviewer #2: The authors explored a simple approach for filtering out physicochemical liability motifs in antibody sequences with a goal of reducing the false positive rates associated with such motifs. The filtering is based on a check if a given motif occurs in the same position in the germline sequence of the antibody, if a given motif is in the same position in the therapeutic antibodies, and if a given motif is structurally exposed to the solvent. The authors showed that the filtered-out liabilities had a lower false positive rate based on analysis of a previously published experimental dataset. Overall, the paper provides a potentially effective approach of reducing false positives rates when evaluating antibody sequence for chemical liabilities, and additionally the authors provide a valuable analysis of chemical liability motif occurrence in a variety of antibody sequence data in general. As such, this work and associated web service will likely be useful for practicing antibody engineers. A possible criticism of the paper is that in terms of conceptual or biological advances, it doesn’t break any new ground and the methodology is quite simple, and mostly based upon published research.

The manuscript could benefit from some revisions outlined below:

1. The description of the germline flag does not indicate how the CDR3 region is handled. Is this region excluded when applying the flag since it is lacking in the germline sequence as it arises from recombination? If it is excluded and a significant fraction of liabilities arise from CDR3, perhaps the presence of the motif in "germline" sequences would be better handled through its frequency in the natural antibody space, similar to the therapeutic flag used in this study. Alternatively, a machine learning approach that accounts for the context may be more appropriate (see below).

2. Using the occurrence of a given motif at a given position within the natural and therapeutics antibody sequence space is predictive of whether a given motif will be a true chemical liability. However, the probability of occurrence for a given amino glosses over other factors, such as the identities of neighbouring amino acids in sequence and in structure. A machine learning method that could account for the context of a given position within the sequence, such as a sequence-based attention model, could potentially be more powerful in evaluating the presence of a given sequence within a larger sequence space, such as the natural antibody space.

3. The manuscript would benefit from more detailed descriptions of the figures and tables, and a more detailed description of the flags in the core of the paper in addition to the methods section.

**Have the authors made all data and (if applicable) computational code underlying the findings in their manuscript fully available?**

Reviewer #1: Yes

Reviewer #2: Yes

PLOS authors have the option to publish the peer review history of their article (what does this mean?). If published, this will include your full peer review and any attached files.

Reviewer #1: **Yes: **Camille Bilodeau

Reviewer #2: No
---

## [Editor Report · Decision Letter 1]

1 Feb 2024

Dear Dr Krawczyk,

We are pleased to inform you that your manuscript 'LAP: Liability Antibody Profiler by sequence & structural mapping of natural and therapeutic antibodies' has been provisionally accepted for publication in PLOS Computational Biology.

Best regards,

Nir Ben-Tal

Section Editor

PLOS Computational Biology

---

## [Editor Report · Acceptance letter]

29 Feb 2024

PCOMPBIOL-D-23-01696R1 

LAP: Liability Antibody Profiler by sequence & structural mapping of natural and therapeutic antibodies

Dear Dr Krawczyk,

I am pleased to inform you that your manuscript has been formally accepted for publication in PLOS Computational Biology. Your manuscript is now with our production department and you will be notified of the publication date in due course.

With kind regards,

Zsofia Freund
